# AdaDGS: An adaptive black-box optimization method with a nonlocal directional Gaussian smoothing gradient

## Abstract

The local gradient points to the direction of the steepest slope in an *infinitesimal* neighborhood. An optimizer guided by the local gradient is often trapped in local optima when the loss landscape is multi-modal. A directional Gaussian smoothing (DGS) approach was recently proposed in (Zhang et al., 2020) and used to define a truly *nonlocal* gradient, referred to as *the DGS gradient*, for high-dimensional black-box optimization. Promising results show that replacing the traditional local gradient with the DGS gradient can significantly improve the performance of gradient-based methods in optimizing highly multi-modal loss functions. However, the optimal performance of the DGS gradient may rely on fine tuning of two important hyper-parameters, i.e., the smoothing radius and the learning rate. In this paper, we present a simple, yet ingenious and efficient adaptive approach for optimization with the DGS gradient, which removes the need of hyper-parameter fine tuning. Since the DGS gradient generally points to a good search direction, we perform a line search along the DGS direction to determine the step size at each iteration. The learned step size in turn will inform us of the scale of function landscape in the surrounding area, based on which we adjust the smoothing radius accordingly for the next iteration. We present experimental results on high-dimensional benchmark functions, an airfoil design problem and a game content generation problem. The AdaDGS method has shown superior performance over several the state-of-the-art black-box optimization methods.

## 1 Introduction

We consider the problem of black-box optimization, where we search for the optima of a loss function $F : \mathbb{R}^d \to \mathbb{R}$ given access to only its function queries. This type of optimization finds applications in many machine learning areas where the loss function's gradient is inaccessible, or unuseful, for example, in optimizing neural network architecture (Real et al., 2017), reinforcement learning (Salimans et al., 2017), design of adversarial attacks (Chen et al., 2017), and searching the latent space of a generative model (Sinay et al., 2020).

The local gradient, i.e., $\nabla F(\boldsymbol{x})$, is the most commonly used quantities to guide optimization. When $\nabla F(\boldsymbol{x})$ is inaccessible, we usually reformulate $\nabla F(\boldsymbol{x})$ as a functional of $F(\boldsymbol{x})$. One class of methods for reformulation is Gaussian smoothing (GS) (Salimans et al., 2017; Liu et al., 2017; Mania et al., 2018). GS first smooths the loss landscape with $d$-dimensional Gaussian convolution and represents $\nabla F(\boldsymbol{x})$ by the gradient of the smoothed function. Monte Carlo (MC) sampling is used to estimate the Gaussian convolution. It is known that the local gradient $\nabla F(\boldsymbol{x})$ points to the direction of the steepest slope in an *infinitesimal* neighborhood around the current state $\boldsymbol{x}$. An optimizer guided by the local gradient is often trapped in local optima when the loss landscape is non-convex or multi-modal. Despite the improvements (Maggiar et al., 2018; Choromanski et al., 2018; 2019; Sener & Koltun, 2020; Maheswaranathan et al., 2019; Meier et al., 2019), GS did not address the challenge of applying the local gradient to global optimization, especially in high-dimensional spaces.

The nonlocal Directional Gaussian Smoothing (DGS) gradient, originally developed in (Zhang et al., 2020), shows strong potential to alleviate such challenge. The key idea of the DGS gradient is to conduct 1D nonlocal explorations along $d$ orthogonal directions in $\mathbb{R}^d$, each of which defines a non-

local directional derivative as a 1D integral. Then, the $d$ directional derivatives are assembled to form the DGS gradient. Compared with the traditional GS approach, the DGS gradient can use large smoothing radius to achieve *long-range* exploration along the orthogonal directions This enables the DGS gradient to provide better search directions than the local gradient, making it particularly suitable for optimizing multi-modal functions. However, the optimal performance of the DGS gradient may rely on fine tuning of two important hyper-parameters, i.e., the smoothing radius and the learning rate, which limits its applicability in practice.

In this work, we propose *AdaDGS*, an adaptive optimization method based on the DGS gradient. Instead of designing a schedule for updating the learning rate and the smoothing radius as in (Zhang et al., 2020), we learn their update rules automatically from a backtracking line search (Nocedal & Wright, 2006). Our algorithm is based on a simple observation: while the DGS gradient generally points to a good search direction, the best candidate solution along that direction may not locate in nearby neighborhood. More importantly, relying on a single candidate in the search direction based on a prescribed learning rate is simply too susceptible to highly fluctuating landscapes. Therefore, we allow the optimizer to perform more thorough search along the DGS gradient and let the line search determine the step size for the best improvement possible. Our experiments show that the introduction of the line search into the DGS setting requires a small but well-worth extra amount of function queries per iteration. After each line search, we update the smoothing radius according to the learned step size, because this quantity now represents an estimate of the distance to an important mode of the loss function, which we retain in the smoothing process. The performance and comparison of AdaDGS to other methods are demonstrated herein through three medium- and high-dimensional test problems, in particular, a high-dimensional benchmark test suite, an airfoil design problem and a level generation problem for Super Mario Bros.

**Related works.** The literature on black-box optimization is extensive. We only review methods closely related to this work (see (Rios & Sahinidis, 2009; Larson et al., 2019) for overviews).

*Random search*. These methods randomly generate the search direction and either estimate the directional derivative using GS formula or perform direct search for the next candidates. Examples are two-point approaches (Flaxman et al., 2005; Nesterov & Spokoiny, 2017; Duchi et al., 2015; Bubeck & Cesa-Bianchi, 2012), three-point approaches (Bergou et al., 2019), coordinate-descent algorithms (Jamieson et al., 2012), and binary search with adaptive radius (Golovin et al., 2020).

*Zeroth order methods based on local gradient surrogate*. This family mimics first-order methods but approximate the gradient via function queries (Liu et al., 2017; Chen et al., 2019; Balasubramanian & Ghadimi, 2018). A exemplary type of these methods is the particular class of Evolution Strategy (ES) based on the traditional GS, first developed by (Salimans et al., 2017). MC is overwhelmingly used for gradient approximation, and strategies for enhancing MC estimators is an active area of research, see, e.g., (Maggiar et al., 2018; Rowland et al., 2018; Maheswaranathan et al., 2019; Meier et al., 2019; Sener & Koltun, 2020). Nevertheless, these effort only focus on local regime, rather than the nonlocal regime considered in this work.

*Orthogonal exploration*. It has been investigated in black-box optimization, e.g., finite difference explores orthogonal directions. (Choromanski et al., 2018) introduced orthogonal MC sampling into GS for approximating the local gradient; (Zhang et al., 2020) introduced orthogonal exploration and the Gauss-Hermite quadrature to define and approximate a nonlocal gradient.

*Adaptive methods*. Another adaptive method based on DGS gradient can be found in (Dereventsov et al., 2020). Our work is dramatically different in that our update rule for the learning rate and smoothing radius is drawn from line search instead of from Lipschitz constant estimation. The long-range line search can better exploit the DGS direction and thus significantly reduce the number of function evaluations and iterations. Line search is a classical method for selecting learning rate (Nocedal & Wright, 2006) and has also been used in adaptation of some nonlocal search techniques, see, e.g., (Hansen, 2008). In this work, we apply backtracking line search on DGS direction. We do not employ popular terminate conditions such as Armijo (Armijo, 1966) and Wolfe condition (Wolfe, 1969) and always conduct the full line search, as this requires a small extra cost, compared to high-dimensional searching.

## 2    THE DIRECTIONAL GAUSSIAN SMOOTHING (DGS) GRADIENT

We are concerned with solving the following optimization problem

$$\min_{\boldsymbol{x} \in \mathbb{R}^d} F(\boldsymbol{x}),$$

where $\boldsymbol{x} = (x_1, \ldots, x_d) \in \mathbb{R}^d$ consists of $d$ parameters, and $F : \mathbb{R}^d \to \mathbb{R}$ is a $d$-dimensional loss function. The traditional GS method defines the smoothed loss function as $F_\sigma(\boldsymbol{x}) = \mathbb{E}_{\boldsymbol{u} \sim \mathcal{N}(0, \mathbf{I}_d)} [F(\boldsymbol{x} + \sigma \boldsymbol{u})]$, where $\mathcal{N}(0, \mathbf{I}_d)$ is the $d$-dimensional standard Gaussian distribution, and $\sigma > 0$ is the smoothing radius. When the local gradient $\nabla F(\boldsymbol{x})$ is unavailable, the traditional GS uses $\nabla F_\sigma(\boldsymbol{x}) = \frac{1}{\sigma} \mathbb{E}_{\boldsymbol{u} \sim \mathcal{N}(0, \mathbf{I}_d)} [F(\boldsymbol{x} + \sigma \boldsymbol{u}) \boldsymbol{u}]$ (Flaxman et al., 2005) to approximate $\nabla F$ by exploiting $\lim_{\sigma \to 0} \nabla F_\sigma(\boldsymbol{x}) = \nabla F(\boldsymbol{x})$ (i.e., setting $\sigma$ small). Hence, the traditional GS is unsuitable for defining a nonlocal gradient where a large smoothing radius $\sigma$ is needed.

In (Zhang et al., 2020), the DGS gradient was proposed to circumvent this hurdle. The key idea was to apply the 1D Gaussian smoothing along $d$ orthogonal directions, so that only 1D numerical integration is needed. In particular, define a 1D cross section of $F(\boldsymbol{x})$ $G(y \,|\, \boldsymbol{x}, \boldsymbol{\xi}) = F(\boldsymbol{x} + y\,\boldsymbol{\xi}), \;\; y \in \mathbb{R}$, where $\boldsymbol{x}$ is the current state of $F$ and $\boldsymbol{\xi}$ is a unit vector in $\mathbb{R}^d$. Then, the Gaussian smoothing of $F(\boldsymbol{x})$ along $\boldsymbol{\xi}$ is represented as $G_\sigma(y \,|\, \boldsymbol{x}, \boldsymbol{\xi}) := (1/\sqrt{2\pi}) \int_{\mathbb{R}} G(y + \sigma v \,|\, \boldsymbol{x}, \boldsymbol{\xi}) \exp(-v^2/2) dv$. The derivative of the smoothed $F(\boldsymbol{x})$ along $\boldsymbol{\xi}$ is a 1D expectation

$$\mathscr{D}[G_\sigma(0 \,|\, \boldsymbol{x}, \boldsymbol{\xi})] = \frac{1}{\sigma} \, \mathbb{E}_{v \sim \mathcal{N}(0,1)} [G(\sigma v \,|\, \boldsymbol{x}, \boldsymbol{\xi}) \, v],$$

where $\mathscr{D}[\cdot]$ denotes the differential operator. Intuitively, the DGS gradient is formed by assembling these directional derivatives on $d$ orthogonal directions. Let $\boldsymbol{\Xi} := (\boldsymbol{\xi}_1, \ldots, \boldsymbol{\xi}_d)$ be an orthonormal system, the DGS gradient is defined as

$$\nabla_{\sigma, \boldsymbol{\Xi}}[F](\boldsymbol{x}) := \Big[ \mathscr{D}[G_\sigma(0 \,|\, \boldsymbol{x}, \boldsymbol{\xi}_1)], \cdots, \mathscr{D}[G_\sigma(0 \,|\, \boldsymbol{x}, \boldsymbol{\xi}_d)] \Big] \boldsymbol{\Xi},$$

where $\boldsymbol{\Xi}$ and $\sigma$ can be adjusted during an optimization process.

Since each component of $\nabla_{\sigma, \boldsymbol{\Xi}}[F](\boldsymbol{x})$ only involves a 1D integral, (Zhang et al., 2020) proposed to use the Gauss-Hermite (GH) quadrature rule (Abramowitz & Stegun, 1972), where each component $\mathscr{D}[G_\sigma(0 \,|\, \boldsymbol{x}, \boldsymbol{\xi})$ is approximated as

$$\mathscr{D}^M[G_\sigma(0 \,|\, \boldsymbol{x}, \boldsymbol{\xi})] = \frac{1}{\sqrt{\pi}\sigma} \sum_{m=1}^{M} w_m F(\boldsymbol{x} + \sqrt{2}\sigma v_m \boldsymbol{\xi}) \sqrt{2} v_m. \tag{1}$$

Here $\{v_m\}_{m=1}^M$ are the roots of the $M$-th order Hermite polynomial and $\{w_m\}_{m=1}^M$ are quadrature weights, the values of which can be found in (Abramowitz & Stegun, 1972). It was theoretically proved in (Abramowitz & Stegun, 1972) that the error of the GH estimator is $\sim M!/(2^M(2M)!)$ that is much smaller than the MC's error $\sim 1/\sqrt{M}$. Applying the GH quadrature to each component of $\nabla_{\sigma, \boldsymbol{\Xi}}[F](\boldsymbol{x})$, the following estimator is defined for the DGS gradient:

$$\nabla_{\sigma, \boldsymbol{\Xi}}^M[F](\boldsymbol{x}) = \Big[ \mathscr{D}^M[G_\sigma(0 \,|\, \boldsymbol{x}, \boldsymbol{\xi}_1)], \cdots, \mathscr{D}^M[G_\sigma(0 \,|\, \boldsymbol{x}, \boldsymbol{\xi}_d)] \Big] \boldsymbol{\Xi}. \tag{2}$$

Then, the DGS gradient is readily integrated to first-order schemes to replace the local gradient.

## 3    THE ADADGS ALGORITHM

In this section, we describe an adaptive procedure to remove manually designing and tuning the update schedules for the learning rate and the smoothing radius of the DGS-based gradient descent (Zhang et al., 2020). Our intuitions are: (i) for multimodal landscapes, choosing one candidate solution along the search direction according to a *single* learning rate may make insufficient progress, and (ii) the optimal step size, if known, is a good indicator for the width of optima that dominates the surrounding area and could be used to inform smoothing radius update. Following this rationale, AdaDGS first uses backtracking line search to estimate the optimal learning rate, and then uses the acquired step size to update the smoothing radius. AdaDGS is straightforward to implement and we

find this strategy to overcome the sensitivity to the hyper-parameter selection that affects the original DGS method. As we shall see, the most important hyperparameters in AdaDGS control how aggressive we want to conduct the line search. Our key advantage in high-dimensional optimization is that with a modest budget for line search (compared to that for computing DGS gradient), we can still get a very generous number of function evaluations along DGS direction and approximate the optimal learning rate. We suggested some default values of these hyperparameters which are proven to be universally good throughout our test. However, if one prefers, they can definitely adjust these for a more aggressive line search. For example, even doubling or tripling the number of points to be visited along DGS direction will increase the total number of function evaluations by a small fraction (5% and 10% correspondingly).

Recall the gradient descent scheme with DGS

$$\boldsymbol{x}_{t+1} = \boldsymbol{x}_t - \lambda_t \nabla^M_{\sigma,\boldsymbol{\Xi}}[F](\boldsymbol{x}_t),$$

where $\boldsymbol{x}_t$ and $\boldsymbol{x}_{t+1}$ are the candidate solutions at iteration $t$ and $t+1$, and $\lambda_t$ is the learning rate. The details of the AdaDGS algorithm is described below.

*Learning rate update via line search.* At iteration $t$, we perform the line search along $\nabla^M_{\sigma,\boldsymbol{\Xi}}[F](\boldsymbol{x}_t)$ within the interval $[\boldsymbol{x}_t - L_{\min} \frac{\nabla^M_{\sigma,\boldsymbol{\Xi}}[F](\boldsymbol{x}_t)}{\|\nabla^M_{\sigma,\boldsymbol{\Xi}}[F](\boldsymbol{x}_t)\|}, \boldsymbol{x}_t - L_{\max} \frac{\nabla^M_{\sigma,\boldsymbol{\Xi}}[F](\boldsymbol{x}_t)}{\|\nabla^M_{\sigma,\boldsymbol{\Xi}}[F](\boldsymbol{x}_t)\|}]$, where $L_{\max}$ and $L_{\min}$ are the maximum and minimum exploration distances, respectively. We visit $S$ points in the interval, equally spaced on a log scale, and choose the best candidate. The corresponding contraction factor is $\rho = \min\{0.9, (L_{\min}/L_{\max})^{1/(S-1)}\}$. More rigorously, the selected learning rate is

$$\lambda_t := \frac{L_{\max}\rho^J}{\|\nabla^M_{\sigma,\boldsymbol{\Xi}}[F](\boldsymbol{x}_t)\|}, \quad \text{where} \quad J = \underset{j \in \{0,\ldots,S-1\}}{\arg\min} F\left(\boldsymbol{x}_t - L_{\max}\rho^j \frac{\nabla^M_{\sigma,\boldsymbol{\Xi}}[F](\boldsymbol{x}_t)}{\|\nabla^M_{\sigma,\boldsymbol{\Xi}}[F](\boldsymbol{x}_t)\|}\right). \quad (3)$$

The default value of $L_{\max}$ is the length of the diagonal of the search domain. This value could be refined by running some test iterations, but our algorithm is not sensitive to such refining. The default value of $L_{\min}$ is $L_{\min} = 0.005 L_{\max}$. The default value for $S$ is $S = \max\{12, 0.05Md\}$, where $Md$ is the number of samples required by the DGS gradient.

This means that when $d$ is high, we spend roughly 5% budget of function evaluations for line search. Note that when $S$ is large, $\rho = 0.9$ and the actual minimum exploration distance is $L_{\max}0.9^{S-1} < L_{\min}$. As long as the DGS gradient points to a good search direction, the line search along a 1D ray is much more cost-effective than searching in $d$-dimensional spaces.

*Smoothing radius update.* The smoothing radius $\sigma_t$ is adjusted based on the learning rate learned from the line search. The initial radius $\sigma_0$ is set to be on the same scale as the width of the search domain. At iteration $t$, we set $\sigma_t$ to be the mean of the smoothing radius and the learning rate from iteration $t-1$, i.e.,

$$\sigma_t = \frac{1}{2}(\sigma_{t-1} + \lambda_{t-1}). \quad (4)$$

because both quantities indicate the landscape of the loss function.

*The number of Gaussian-Hermite points.* The AdaDGS method is not sensitive to the number of GH points. We do not observe significant benefit of using more than 5 GH quadrature points per direction. In some tests (Section 4.3), 3 GH quadrature points per direction are sufficient.

---

**Algorithm 1**: The AdaDGS algorithm

---

1: **Hyper-parameters**:
   $M$: # GH quadrature points
   $L_{\max}$: the maximum exploration
   $L_{\min}$: the minimum exploration
   $S$: # function eval. per line search
   $\sigma_0$: initial smoothing radius
   $\gamma$: tolerance for triggering random exploration
2: **Input**: The initial state $\boldsymbol{x}_0$
3: **Output**: The final state $\boldsymbol{x}_T$
4: Set $\boldsymbol{\Xi} = \mathbf{I}_d$ (or a random orthonormal matrix)
5: **for** $t = 0, \ldots T - 1$ **do**
6:    Evaluate $\{G(\sqrt{2}\sigma_i v_m \,|\, \boldsymbol{x}_t, \boldsymbol{\xi}_i)\}_{m=1,\ldots,M}^{i=1,\ldots,d}$
7:    **for** $i = 1, \ldots, d$ **do**
8:       Compute $\mathscr{D}^M[G_{\sigma_i}(0 \,|\, \boldsymbol{x}_t, \boldsymbol{\xi}_i)]$ in Eq. (1)
9:    **end for**
10:   Assemble $\nabla^M_{\sigma,\boldsymbol{\Xi}}[F](\boldsymbol{x}_t)$ in Eq. (2)
11:   Update $\lambda_t$ according to Eq. (3)
12:   Set $\boldsymbol{x}_{t+1} = \boldsymbol{x}_t - \lambda_t \nabla^M_{\sigma,\boldsymbol{\Xi}}[F](\boldsymbol{x}_t)$
13:   Set $\sigma_{t+1} = \frac{1}{2}(\sigma_t + \lambda_t)$ according to Eq. (4)
14:   **if** $|F(\boldsymbol{x}_t) - F(\boldsymbol{x}_{t-1})|/|F(\boldsymbol{x}_{t-1})| < \gamma$ **then**
15:      Generate a random rotation $\boldsymbol{\Xi}$
16:      Set $\sigma_{t+1} = \sigma_0$
17:   **end if**
18: **end for**

---

*Random exploration.* We incorporate the following strategies to support random exploration and help the AdaDGS algorithm escape undesirable scenarios. We use the condition $|F(\boldsymbol{x}_t) - F(\boldsymbol{x}_{t-1})|/|F(\boldsymbol{x}_{t-1})| < \gamma$ to trigger the random exploration, where the default value for $\gamma$ is 0.001. Users can optionally trigger these strategies when the method fails to make progress, e.g., insufficient decrease or too small step size.

- *Reset the smoothing radius.* Since $\sigma$ is updated following Eq. (4), $\sigma$ becomes small with the learning rate. Thus, we occasionally reset $\sigma$ to its initial value. We set a minimum interval of 10 iterations between two consecutive resets. In many of our tests, the function values reached by AdaDGS within 10 first iterations (before the radius reset is triggered) are already lower than those by its competitors can at the end.

- *Random generation of $\Xi$.* Keeping the directional smoothing along a fixed set of coordinates $\Xi$ may eventually reduce the exploration capability. To alleviate this issue, we occasionally change the nonlocal exploration directions by randomly generating an orthogonal matrix $\Xi$. An important difference between our approach and the random perturbation strategy in (Zhang et al., 2020) is that the approach in (Zhang et al., 2020) only add small perturbation to the identity matrix, but we generate a totally random rotation matrix.

## 4 EXPERIMENTS

We present the experimental results using three sets of problems. All experiments were implemented in Python 3.6 and conducted on a set of cloud servers with Intel Xeon E5 CPUs.

### 4.1 TESTS ON HIGH-DIMENSIONAL BENCHMARK FUNCTIONS

We compare the AdaDGS method with the following (a) DGS: the baseline DGS with polynomial decay update schedule developed in (Zhang et al., 2020), (b) ES-Bpop: the standard OpenAI evolution strategy in (Salimans et al., 2017) with a big population (i.e., using the same number of samples as AdaDGS), (c) ASEBO : Adaptive ES-Active Subspaces for Blackbox Optimization (Choromanski et al., 2019) with a population of size $4 + 3\log(d)$, (d) IPop-CMA: the restart covariance matrix adaptation evolution strategy with increased population size (Auger & Hansen, 2005), (e) Nesterov: the random search method in (Nesterov & Spokoiny, 2017), (f) FD: the classical central difference scheme, and (g) TuRBO: trust region Bayesian optimization (Eriksson et al., 2019). The information of the codes used for the baselines is provided in Appendix.

We test the performance of the AdaDGS method on 12 high-dimensional benchmark functions (El-Abd, 2010; Jamil & Yang, 2013), including $F_1(\boldsymbol{x})$: Ackley, $F_2(\boldsymbol{x})$: Alpine, $F_3(\boldsymbol{x})$: Ellipsoidal, $F_4(\boldsymbol{x})$: Quintic, $F_5(\boldsymbol{x})$: Rastrigin, $F_6(\boldsymbol{x})$: Rosenbrock, $F_7(\boldsymbol{x})$: Salomon, $F_8(\boldsymbol{x})$: Schaffer's F7, $F_9(\boldsymbol{x})$: Sharp-Ridge, $F_{10}(\boldsymbol{x})$: Sphere, $F_{11}(\boldsymbol{x})$: Trigonometric, and $F_{12}(\boldsymbol{x})$: Wavy. To make the test functions more general, we applied the following linear transformation to $\boldsymbol{x}$,

$$\boldsymbol{z} = \mathbf{R}(\boldsymbol{x} + \boldsymbol{x}_{\text{opt}} - \boldsymbol{x}_{\text{loc}}),$$

which first moves the optimal state $\boldsymbol{x}_{\text{opt}}$ to a new random location $\boldsymbol{x}_{\text{loc}}$ and then applies a random rotation $\mathbf{R}$ to make the function non-separable. We substitute $\boldsymbol{z}$ into the standard definitions of the benchmark functions to formulate our test problems. Details about those functions are provided in Appendix.

The hyper-parameters of the AdaDGS method are fixed for the six test functions. Specifically, $L_{\max}$ is the length of the diagonal of the domain, $S = 200 \ (= 0.05Md)$, $\sigma_0 \sim 5 * \text{width}$, and $M = 5$. Since $S$ is large, the minimum exploration distance is easily small and we do not need to concerned with $L_{\min}$. We choose contraction factor to be 0.9. We turned off the random perturbation by setting $\gamma = 0$. For each test function, we performed 20 trials, each of which has a random initial state, a random rotation matrix $\mathbf{R}$ and a random location of $\boldsymbol{x}_{\text{loc}}$.

The comparison between AdaDGS and the baselines in the 1000D case are shown in Figure 1. Additional results are shown in Appendix C, where the loss decay is plotted in log-scale. The AdaDGS has the best performance overall. In particular, the improvement of AdaDGS over baseline DGS is significant, demonstrating the effectiveness of our adaptive mechanism. AdaDGS shows substantially superior performance in optimizing the highly multimodal functions $F_1$, $F_2$, $F_4$, $F_5$, $F_7$, $F_8$, $F_{11}$, which is significant in global optimization. For the ill-conditioned functions $F_3$, $F_6$

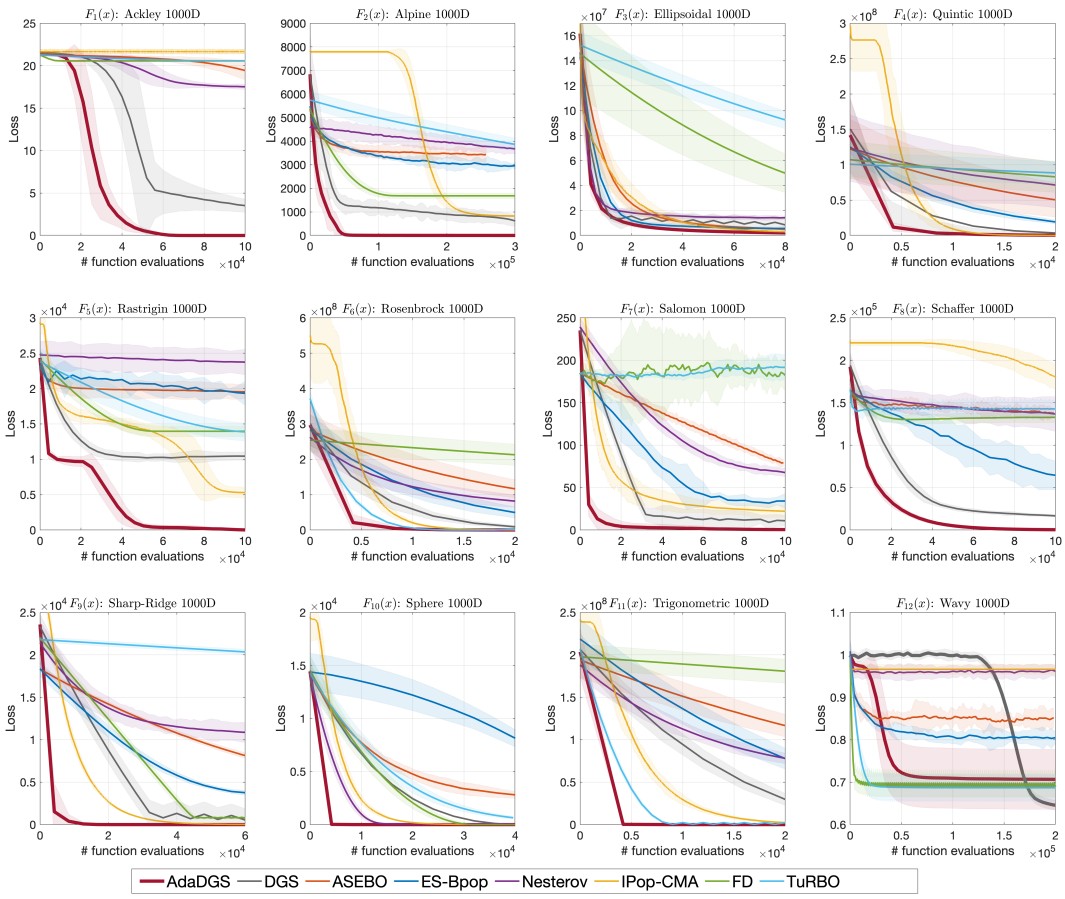

Figure 1: Comparison of the loss decay w.r.t. # function evaluations for the 12 benchmark functions in 1000D. Each curve is the mean of 20 independent trials and the shaded areas represent [mean-3std, mean+3std]. The global minimum is $F_i(\boldsymbol{x}_{\text{opt}}) = 0$ except for $i = 11$, where $F_{11}(\boldsymbol{x}_{\text{opt}}) = 1$. The AdaDGS has the best performance overall, especially for the highly multi-modal functions $F_1$, $F_2$, $F_4$, $F_5$, $F_7$, $F_8$, $F_{11}$, $F_{11}$. All the methods fail to find the global minimum of $F_{12}$ which has no global structure to exploit.

and $F_9$, AdaDGS can at least match the performance of the best baseline method, e.g., IPop-CMA. The test with sphere function $F_{10}$ show AdaDGS converges within 2 steps, confirming the quality of DGS search direction. For $F_{12}$, all the methods fail to find the global minimum because it is highly multi-modal and there is no global structure to exploit, which makes it extremely challenging for all global optimization methods. We also tested AdaDGS in 2000D, 4000D and 6000D to illustrate its scalability with the dimension. The hyper-parameters are set the same as the 1000D cases. The results are shown in Figure 2. The AdaDGS method still achieves promising performance, even though the number of total function evaluations increases with the dimension.

## 4.2 TESTS ON AIRFOIL SHAPE OPTIMIZATION

We applied the AdaDGS method to design a 2D airfoil. We used a computational fluid dynamics (CFD) code, XFoil v.6.91 (Drela, 1989), and its Python interface v.1.1.1[1]. The Xfoil can conduct CFD simulations of an airfoil given a 2D contour design. The first step is to choose an appropriate parameterization for the upper and lower parts of the airfoil. In this work, we used the state-of-the-art Class/Shape function Transformation (CST) (Kulfan, 2008). Specifically, the upper/down airfoil geometry is represented as $z(x) = \sqrt{x}(1 - x)\Sigma_{i=0}^{N}[A_i\binom{N}{i}x^i(1-x)^{N-i}] + x\Delta z_{\text{te}}$, where $x \in [0, 1]$, $N$ is the polynomial order. The polynomial

---

[1] Avaliable at `https://github.com/daniel-de-vries/xfoil-python`.

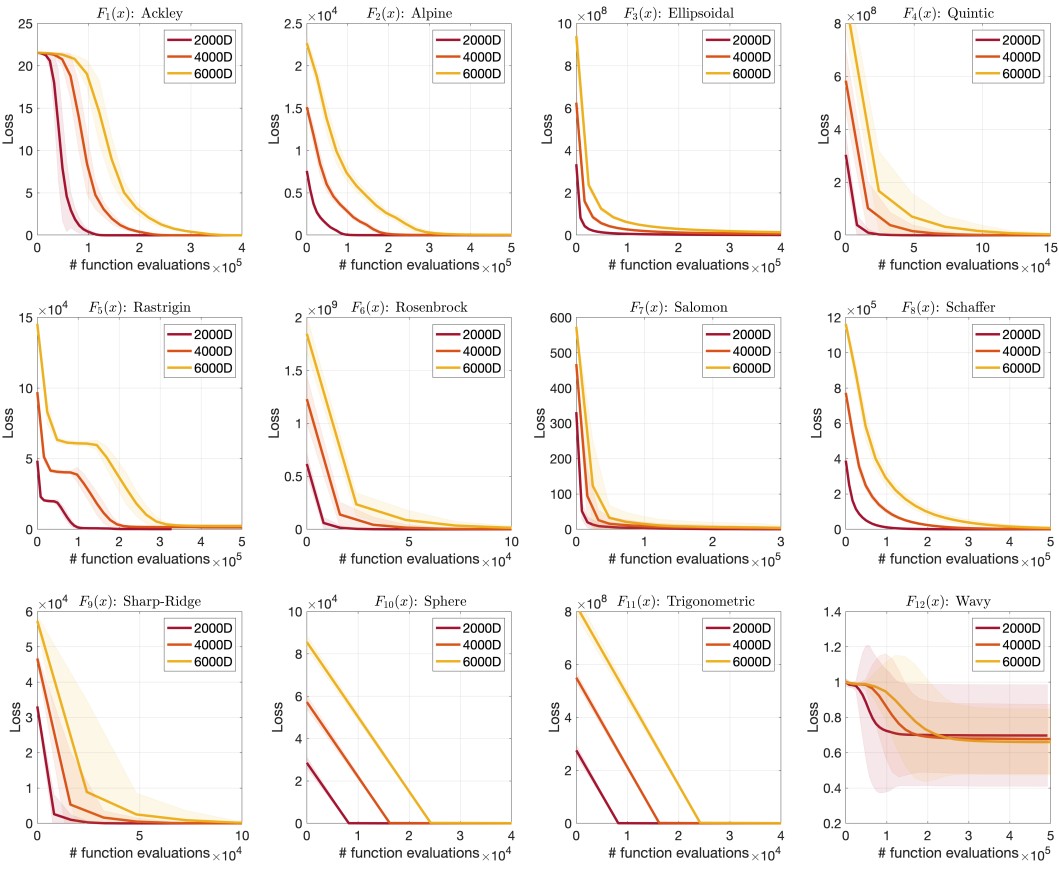

Figure 2: Tests on AdaDGS's scalability in 2000D, 4000D and 6000D. The hyper-parameters are the same as the 1000D case. The AdaDGS still achieves promising performance, even though the number of total function evaluations increases with the dimension.

coefficients $A_i$ and the position of the airfoil tail $\Delta z_{te}$ are the parameters needed to be optimized. We used two different CST polynomials to parameterize the upper and lower part of the airfoil, where the polynomial degree for each polynomial is set to 6 by following the suggestion in (Ceze et al.). Then, the dimension of the optimization problem is $d = 15$. The initial search domain is set to $[-1, 1]^d$. We simulated all models with Reynolds number 12e6, speed 0.4 mach and the angles of attack from 5 to 8 degrees. The initial condition is the standard NACA 0012 AIR-

Table 1: The airfoil lift and drag values after 1500 calls to XFoil. AdaDGS provides the best design with the biggest Lift-Drag.

|  | Init | The best design after 1500 simulations | | |
|---|---|---|---|---|
|  |  | AdaDGS | ASEBO | ES-Bpop |
| Lift | 0.0473 | **2.6910** | 1.2904 | 1.1931 |
| Drag | 0.0161 | 0.0133 | 0.0097 | 0.0100 |
| Lift-Drag | 0.0312 | **2.6777** | 1.2821 | 1.1831 |
| Foil |  |  |  |  |

|  | The best design after 1500 simulations | | | |
|---|---|---|---|---|
|  | Nesterov | Ipop-CMA | FD | TuRBO |
| Lift | 0.8575 | 1.1403 | 0.8606 | 2.0747 |
| Drag | 0.0095 | 0.0072 | **0.0071** | 0.0097 |
| Lift-Drag | 0.8480 | 1.1331 | 0.8535 | 2.0650 |
| Foil |  |  |  |  |

FOIL. The hyper-parameters of the AdaDGS method are $L_{max}$ is the length of the diagonal of the domain, $L_{min} = 0.005 L_{max}$, $S = 12$, $\sigma_0 =$ search domain width, $M = 5$ and $\gamma = 0.001$. The gain function is set to Lift-Drag and the goal is to maximize the gain. The results are shown in Table 1. With 1500 simulations, all the methods reach a shape with Lift > Drag, which means the airfoils can fly under the experimental scenario. Our AdaDGS method produced the best design, i.e., biggest

Lift-Drag. The other baselines achieved lower Drag than the AdaDGS but did not achieve very high Lift force.

### 4.3 TESTS ON GAME CONTENT GENERATION FOR SUPER MARIO BROS

We apply the AdaDGS method to generate a variety of Mario game levels with desired attributes. These levels are produced by generative adversarial networks (GAN) (Goodfellow et al., 2014), which map from latent parameters to high-quality images. To generate a game level with desired characteristic, one needs to search in the latent space of the GAN for parameters that optimize a prescribed stylistic or performance metric.

In this paper, we evaluate the performance of AdaDGS in generating game levels for two different types of objectives: (i) levels that have the maximum number of certain tiles. We consider sky tiles (i.e., game objects that lie in the above half of the image) **(MaxSkyTiles)** and enemy tiles **(MaxEnemies)**; (ii) playable levels that require the AI agent perform maximum certain action. We consider jumping **(MaxJumps)** and killing an enemy **(MaxKills)**. These characteristics are often considered for evaluating latent space search and optimization methods (Volz et al., 2018; 2019; Fontaine et al., 2020). Specifically for type (ii) objective, we use the AI agent developed by Robin Baumgarten[2] to evaluate the playability of the level and the objective functions. We set unplayable penalty to be 100 and add that to the objective function when the generated level is unplayable. The game levels are generated from a pre-trained DCGAN by (Fontaine et al., 2020), whose inputs are vectors in $[-1, 1]^{32}$. Details of the architecture can also be found in (Volz et al., 2018).

The hyper-parameters of the AdaDGS method are set at default values for the four tests. Specifically, $L_{\max}$ is the length of the diagonal of the domain, $L_{\min} = 0.029$ ($= 0.005 L_{\max}$), $S = 12$, $\sigma_0 =$ search domain width, $M = 3$ and $\gamma = 0.001$. We start with $\Xi$ being a random orthonormal matrix generated by `scipy.stats.ortho_group.rvs`. As demonstrated in (Volz et al., 2018), the IPop-CMA is by far the mostly used and superior method for this optimization task, so we only compared the performance of our method with IPop-CMA. We used the pycma v.3.0.3 with the population size be 17 and the radius be 0.5, as described in (Fontaine et al., 2020). We apply *tanh* function to the latent variables before sending it to the generator model, because this model was trained on $[-1, 1]^{32}$. 50 trials with random initialization are run for each test.

The comparison between AdaDGS and IPop-CMA are shown in Figure 3. AdaDGS outperforms IPop-CMA in three out of four test functions and is close in the other. We find that Ipop-CMA can also find the best optima in many trials, but it is easier to get stuck at undesirable modes, e.g, local minima. Taking the MaxSkyTiles case as an example. There are 4 types of patterns, shown in Figure 4, are generated by AdaDGS and IPop-CMA in maximizing MaxSkyTiles. The top-left pattern in Figure 4 is the targeted one, and the other three represent different types of local minima. The probability of generating the targeted pattern is 90% for AdaDGS, and 74% for IPop-CMA.

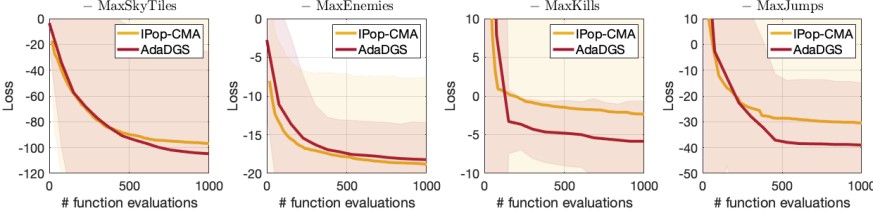

Figure 3: Comparison of the loss decay w.r.t. # function evaluations for four objectives. From left to right: generate a Mario level with i) maximum number of sky tiles, ii) maximum number of enemies, iii) forcing AI Mario to make the most kills, and iv) forcing AI Mario to make the most jumps.

## 5 CONCLUSION

We developed an adaptive optimization algorithm with the DGS gradient, which successfully removed the need of hyper-parameter fine tuning of the original DGS method in (Zhang et al., 2020). Experimental results demonstrated the superior performance of the AdaDGS method compared to

---

[2]https://www.youtube.com/watch?v=DlkMs4ZHHr8

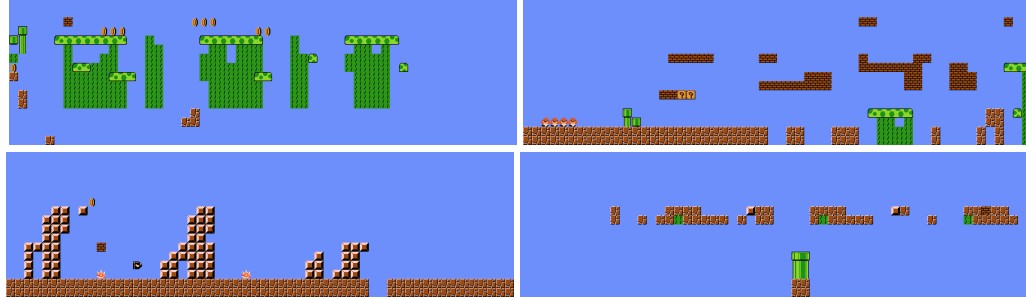

Figure 4: The levels generated by optimizing MaxSkyTiles objective have four patterns. From top left, clockwise: High number (>80) of sky tiles, medium number ($\simeq 40$) of sky tiles, medium number of sky tiles with no ground, and low number ($\simeq 20$) of sky tiles. The top-left type of patterns is the targeted pattern and the other three represent local minima. The probabilities of generating the four types of patterns are: AdaDGS: 90%, 4%, 2%, 4% and IPop-CMA: 74%, 8%, 8%, 10% (from top left, clockwise). AdaDGS shows better performance on generating the targeted pattern.

several the state-of-the-art black-box optimization methods. On the other hand, the AdaDGS method has some drawbacks that need to be addressed. The most important one is *sampling complexity*. The GH quadrature requires $M \times d$ samples per iteration, which is much more than samples required by MC estimators. The reasons why the AdaDGS outperforms several ES-type methods are due to the good quality of the DGS gradient direction and the line search which significantly reduces the number of iterations. However, when the computing budget is very limited (e.g., only allowing $d$ function evaluations for a $d$-dimensional problem), then our method becomes inapplicable. One way to alleviate this challenge is to adopt dimensionality reduction (DR) techniques (Choromanski et al., 2019), such as active subspace and sliced linear regression, and apply the AdaDGS in a subspace to reduce the sampling complexity. Incorporating DR into the AdaDGS method will be considered in our future research.

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

## APPENDIX

## A    DEFINITION OF BENCHMARK FUNCTIONS

Here we provide the definition of the 12 benchmark functions tested in Section 4.1. Let $\Omega$ denote the search domain. To make the test functions more general, we randomly translate the functions so that the optimal state $\boldsymbol{x}_{\text{opt}}$ moves to new location $\boldsymbol{x}_{\text{loc}}$, and then apply a rotation. This transformation can be written as

$$\boldsymbol{z} = \mathbf{R}(\boldsymbol{x} + \boldsymbol{x}_{\text{opt}} - \boldsymbol{x}_{\text{loc}}),$$

where $\mathbf{R}$ is a rotation matrix making the functions non-separable and $\boldsymbol{x}_{\text{loc}}$ is a random location in $0.8\Omega$. The optimization will need to travel farther to reach the new optimal state. We substitute $\boldsymbol{z}$ into the standard definitions of the benchmark functions to formulate our test problems.

- $F_1(\boldsymbol{x})$: **Ackley** function

$$F_1(\boldsymbol{x}) = -a\exp\left(-b\sqrt{\frac{1}{d}\sum_{i=1}^{d}x_i^2}\right) - \exp\left(\frac{1}{d}\sum_{i=1}^{d}\cos(cx_i)\right) + a + \exp(1),$$

where $d$ is the dimension and $a = 20, b = 0.2, c = 2\pi$ are used in our experiments. The initial search domain $\boldsymbol{x} \in [-32.768, 32.768]^d$. The global minimum is $f(\boldsymbol{x}_{\text{opt}}) = 0$ at $\boldsymbol{x}_{\text{opt}} = (0, \ldots, 0)$. The Ackley function represents *non-convex* landscapes with *nearly flat outer region*. The function poses a risk for optimization algorithms, particularly hill-climbing algorithms, to be trapped in one of its many local minima.

- $F_2(\boldsymbol{x})$: **Alpine** function

$$F_2(\boldsymbol{x}) = \sum_{i=1}^{d} |x_i \sin(x_i) + 0.1x_i|,$$

where the initial search domain is $\boldsymbol{x} \in [-10, 10]^d$. The global minimum is $f(\boldsymbol{x}_{\text{opt}}) = 0$, with $8^d$ solutions. We choose $\boldsymbol{x}_{\text{opt}} = (0, \ldots, 0)$. This function represents *multimodal* landscapes with *non-unique global optimum*.

- $F_3(\boldsymbol{x})$: **Ellipsoidal** function

$$F_3(\boldsymbol{x}) = \sum_{i=1}^{d} 10^{6\frac{i-1}{d-1}} x_i^2,$$

where $d$ is the dimension and $\boldsymbol{x} \in [-2, 2]^d$ is the input domain. The global minimum is $f(\boldsymbol{x}_{\text{opt}}) = 0$ at $\boldsymbol{x}_{\text{opt}} = (0, \ldots, 0)$. This represents *convex and highly ill-conditioned* landscapes.

- $F_4(\boldsymbol{x})$: **Quintic** function

$$F_4(\boldsymbol{x}) = \sum_{i=1}^{d} |x_i^5 - 3x_i^4 + 4x_i^3 + 2x_i^2 - 10x_i - 4|,$$

where $\boldsymbol{x} \in [-10, 10]^d$ is the initial search domain. The global minimum is $f(\boldsymbol{x}_{\text{opt}}) = 0$, at $x_i$ is either $-1$ or $2$. We choose $\boldsymbol{x}_{\text{opt}} = (-1, \ldots, -1)$. This function represents *multimodal* landscapes with *global structure*.

- $F_5(\boldsymbol{x})$: **Rastrigin** function

$$F_5(\boldsymbol{x}) = 10d + \sum_{i=1}^{d} [x_i^2 - 10\cos(2\pi x_i)],$$

where $d$ is the dimension and $\boldsymbol{x} \in [-5.12, 5.12]^d$ is the initial search domain. The global minimum is $f(\boldsymbol{x}_{\text{opt}}) = 0$ at $\boldsymbol{x}_{\text{opt}} = (0, \ldots, 0)$. This function represents *highly multimodal* landscapes with *global structure*.

- $F_6(\boldsymbol{x})$: **Rosenbrock** function

$$F_6(\boldsymbol{x}) = \sum_{i=1}^{d-1} [100(x_{i+1} - x_i^2)^2 + (x_i - 1)^2],$$

where $d$ is the dimension and $\boldsymbol{x} \in [-5, 10]^d$ is the initial search domain. The global minimum is $f(\boldsymbol{x}_{\text{opt}}) = 0$ at $\boldsymbol{x}_{\text{opt}} = (1, \ldots, 1)$. The function is *unimodal*, and the global minimum lies in a *bending ridge*, which needs to be followed to reach solution.

- $F_7(\boldsymbol{x})$: **Salomon** function

$$F_7(\boldsymbol{x}) = 1 - \cos\left(2\pi\sqrt{\sum_{i=1}^{d} x_i^2}\right) + 0.1\sqrt{\sum_{i=1}^{d} x_i^2},$$

where $\boldsymbol{x} \in [-100, 100]^d$ is the initial search domain. The global minimum is $f(\boldsymbol{x}_{\text{opt}}) = 0$ at $\boldsymbol{x}_{\text{opt}} = (0, \ldots, 0)$. This function represents *multimodal* landscapes with *global structure*.

- $F_8(\boldsymbol{x})$: **Schaffer** function

$$F_8(\boldsymbol{x}) = \frac{1}{d-1}\left(\sum_{i=1}^{d-1}\left(\sqrt{s_i} + \sqrt{s_i}\sin^2(50s_i^{\frac{1}{5}})\right)\right)^2 \quad \text{with} \quad s_i = \sqrt{x_i^2 + x_{i+1}^2},$$

where $\boldsymbol{x} \in [-100, 100]^d$ is the initial search domain. The global minimum is $f(\boldsymbol{x}_{\text{opt}}) = 0$ at $\boldsymbol{x}_{\text{opt}} = (0, \ldots, 0)$. This function represents *highly multimodal* landscapes.

- $F_9(\boldsymbol{x})$: **Sharp-Ridge** function

$$F_9(\boldsymbol{x}) = x_1^2 + 100\sqrt{\sum_{i=2}^{d} x_i^2},$$

where $d$ is the dimension and $\boldsymbol{x} \in [-10, 10]^d$ is the initial search domain. The global minimum is $f(\boldsymbol{x}_{\text{opt}}) = 0$ at $\boldsymbol{x}_{\text{opt}} = (0, \ldots, 0)$. This represents *convex and anisotropic* landscapes. There is a sharp ridge defined along $x_2^2 + \cdots + x_d^2 = 0$ that must be followed to reach the global minimum, which creates difficulties for optimizations algorithms.

- $F_{10}(\boldsymbol{x})$: **Sphere** function

$$F_{10}(\boldsymbol{x}) = \sum_{i=1}^{d} x_i^2,$$

where $\boldsymbol{x} \in [-5.12, 5.12]^d$ is the initial search domain. The global minimum is $f(\boldsymbol{x}_{\text{opt}}) = 0$, at $\boldsymbol{x}_{\text{opt}} = (0, \cdots, 0)$. The Sphere function represents *unimodal, isotropic* landscapes, and can be used to test the quality of the search direction.

- $F_{11}(\boldsymbol{x})$: **Trigonometric** function

$$F_{11}(\boldsymbol{x}) = 1 + \sum_{i=1}^{d} \{8\sin^2[7(x_i - 0.9)^2] + 6\sin^2[14(x_i - 0.9)^2] + (x_i - 0.9)^2\},$$

where $\boldsymbol{x} \in [-500, 500]^d$ is the initial search domain. The global minimum is $f(\boldsymbol{x}_{\text{opt}}) = 1$, at $\boldsymbol{x}_{\text{opt}} = (0.9, \cdots, 0.9)$. This function represents *multimodal* landscapes with *global structure*.

- $F_{12}(\boldsymbol{x})$: **Wavy** function

$$F_{12}(\boldsymbol{x}) = 1 - \frac{1}{d} \sum_{i=1}^{d} \cos(kx_i) \exp\left(\frac{-x_i^2}{2}\right),$$

where $k = 10$, $\boldsymbol{x} \in [-\pi, \pi]^d$ is the initial domain. The global minimum is $f(\boldsymbol{x}_{\text{opt}}) = 0$, at $\boldsymbol{x}_{\text{opt}} = (0, \cdots, 0)$. This function represents *multimodal* landscapes with *no global structure*.

## B   ADDITIONAL INFORMATION ABOUT THE BASELINE METHODS

### B.1   THE ES-BPOP METHOD

ES-Bpop refers to the standard OpenAI evolution strategy in (Salimans et al., 2017) with the a big population, i.e., the same population size as the AdaDGS method. The purpose of using a big population is to compare the MC-based estimator for the standard GS gradient and the GH-based estimator for the DGS gradient given the same computational cost.

### B.2   THE ASEBO METHOD

ASEBO refers to Adaptive ES-Active Subspaces for Blackbox Optimization proposed in (Choromanski et al., 2019). This is the state-of-the-art method in the family of ES. It has been shown that other recent developments on ES, e.g., (Akimoto & Hansen, 2016; Loshchilov et al., 2019), underperform ASEBO in optimizing the benchmark functions. We use the code published at `https://github.com/jparkerholder/ASEBO` by the authors of the ASEBO method with default hyper-parameters provided in the code.

### B.3   THE IPOP-CMA METHOD

IPop-CMA refers to the restart covariance matrix adaptation evolution strategy with increased population size proposed in (Auger & Hansen, 2005). We use the code pycma v3.0.3 available at `https://github.com/CMA-ES/pycma`. The main subroutine we use is `cma.fmin`, in which the hyper-parameters are

- `restarts=9`: the maximum number of restarts with increasing population size;
- `incpopsize=2`: multiplier for increasing the population size before each restart;
- $\sigma_0$: the initial exploration radius is set to 1/4 of the search domain width.

### B.4   THE NESTEROV METHOD

Nesterov refers to the random search method proposed in (Nesterov & Spokoiny, 2017). We use the stochastic oracle

$$\boldsymbol{x}_{t+1} = \boldsymbol{x}_t - \lambda_t F'(\boldsymbol{x}_t, \boldsymbol{u}_t),$$

where $\boldsymbol{u}_t$ is a randomly selected direction and $F'(\boldsymbol{x}_t, \boldsymbol{u}_t)$ is the directional derivative along $\boldsymbol{u}_t$. According to the analysis in (Nesterov & Spokoiny, 2017), this oracle is more powerful and can be used for non-convex non-smooth functions. As suggested in (Nesterov & Spokoiny, 2017), we use forward difference scheme to compute the directional derivative.

### B.5   THE FD METHOD

FD refers to the classical central difference scheme for local gradient estimation. We implemented our own FD code following the standard numerical recipe.

### B.6 THE TURBO METHOD

TuRBO refers to Trust-Region Bayesian Optimization proposed in (Eriksson et al., 2019), which is the state-of-the-art Bayesian optimization method. We used the code released by the authors at `https://github.com/uber-research/TuRBO`.

## C  ADDITIONAL RESULTS ON HIGH-DIMENSIONAL BENCHCHMARK FUNCTIONS

AdaDGS converges to the global minimum in six of our test functions. Other baselines fail to converge for any of the test functions with the same number of function evaluations. In Figure 5, we plot the performance of AdaDGS and other baselines in log scale to show the convergence of our method. Figure 6 compares the convergence rate of the AdaDGS when changing the dimension of the test functions. With exception of Rastrigin function, the convergence rate does not change when we increase the dimension from 1000 to 6000.

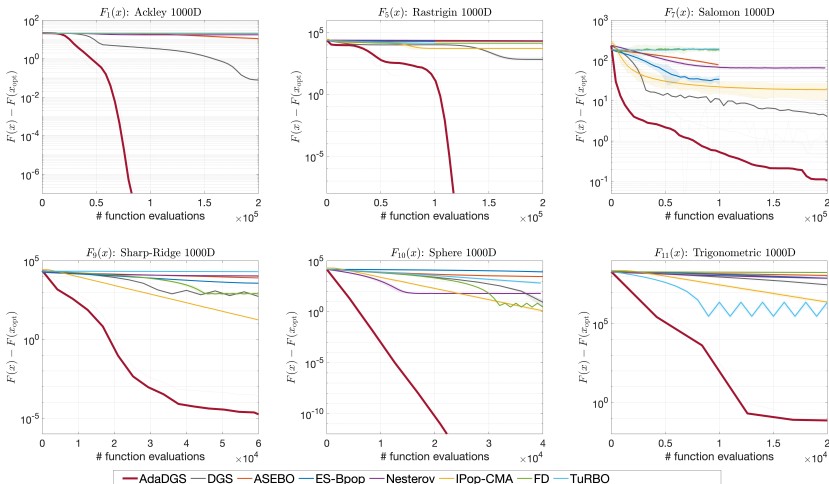

Figure 5: Comparison of the loss decay w.r.t. # function evaluations in log-scale. AdaDGS converges to the global minimum in six of the test functions: $F_1$, $F_5$, $F_7$, $F_9$, $F_{10}$, $F_{11}$. Other baselines fail to converge for any of the test functions. Each curve is the mean of 20 independent trials.

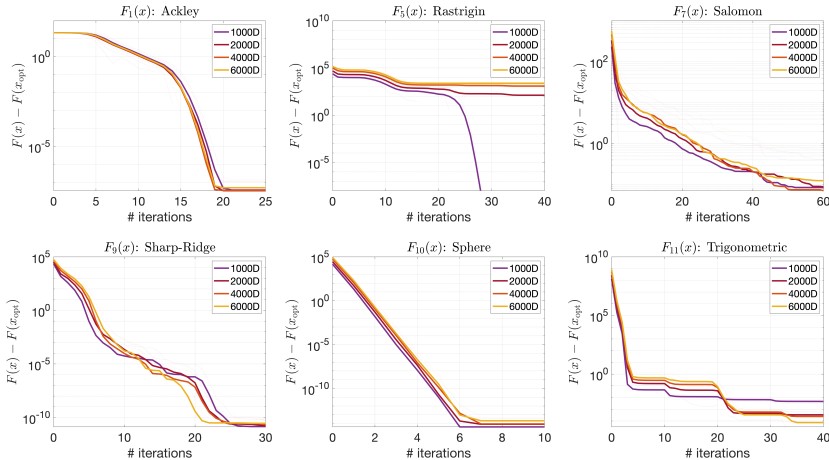

Figure 6: Illustration of the dimension dependence of the convergence rate of AdaDGS method. The convergence rate, i.e., the number of iterations to converge, does not change when we increase the dimension from 1000D to 6000D, with exception of Rastrigin function. All the tests need 60 iterations or less.

