# OpenReview forum: "AdaDGS: An adaptive black-box optimization method with a nonlocal directional Gaussian smoothing gradient"
_ICLR.cc/2021/Conference — Reject_

### Official Review · AnonReviewer3 · 2020-10-13
**Tuning Directional Gaussian Smoothing for global non-convex zeroth order optimization**

**Rating:** 5
**Confidence:** 3

**Review:**

The authors study the problem of global non-convex optimization with access only to function valuations. Specifically, they propose an approach to automatically control the hyper-parameters of Directional Gaussian Smoothing (DGS) a recently proposed solution for the problem. Their proposed solution trade-offs some additional function evaluations per parameter update to perform a line search for the optimal learning rate. Then the tuned learning rate informs the update for the Gaussian smoothing parameter. The proposed automated tuning approach is supported by a large set of experiments that include standard global optimization benchmarks as well as practical applications.

Pros:
1) The authors provide intuition behind the key choices of their automated tuning approach.
2) The experimental results are comprehensive and the proposed approach has (sometimes substantially) better performance than the alternatives.

Cons:
1) The proposed tuning techniques are not very particular to the specifics of the DGS algorithm. Line search for tuning the learning rate is applicable to a wide variety of optimization algorithms. The update rule of Equation (4) could be applicable to any algorithm that employs Gaussian smoothing. It would be helpful if the authors explained how their tuning heuristics are tied to DGS.
2)  The main effort of the authors is to avoid having to use learning rate and smoothing update schedules as they introduce many hyper-parameters. However, the author's tuning approach also introduces many hyper-parameters as well like $L_{\max}$, $L_{\min}$, $\gamma$ and $S$. It would help if the authors explained why the newly introduced parameters can be chosen in a more straightforward fashion compared to the parameters they replace.
3) In the experimental analysis, the authors show that DGS with their tuning approach outperforms the baselines. Given that plain DGS (with some default adaptation schedule) is not part of the baselines, it is not clear if the performance difference should be attributed to DGS or the tuning mechanism. Adding DGS as a baseline would go a long way towards analyzing the contribution of the tuning algorithm.

For now, I am assigning a weak reject score. However, I am willing to substantially increase my score if the authors address the above concerns.

---

> ### Author Response · Authors · 2020-11-18
> **Response to AnonReviewer3**
>
> We thank the reviewer for the constructive comments and suggestions. We address the reviewer’s concerns below.
>
> 1.	**How the tuning heuristic are tied to DGS:** We agree that line search is applicable to a wide variety of optimization algorithms. However, using line search is only beneficial if the search direction is already good. That is exactly what DGS offers – a search direction that supports long-range exploration. Line search then complements DGS with two critical pieces of information: the step size and the smoothing radius.
> Conventional (global) Gaussian smoothing in fact cannot make use of our radius update rule. Due to the heavy requirement on number of function evaluations, this technique is essentially limited to approximate local gradient. The smoothing radius needs to be small, so the tuning heuristic cannot be applied.
>
> 2.	**Hyperparameter selection:** The hyperparameters of our AdaDGS algorithm are indeed chosen intuitively and not sensitive. Please see the general response for our discussion on the selection of new hyperparameters.
>
> 3.	**Adding DGS as a baseline:** The reviewer was absolutely right. We compare the performance of AdaDGS with baseline DGS in the revised paper. The adaptive component makes a clear improvement over non-adaptive DGS.

---

### Official Review · AnonReviewer2 · 2020-10-20
**Too many weak spots, experiments in particular**

**Rating:** 3
**Confidence:** 5

**Review:**

SUMMARY

The paper is heavily based on [Zhang et al., 2020], where the approach of building a "truly non-local gradient" is brought forward. That idea appears in many variants in the literature. As far as I can see, it simply amounts to estimating the gradient from samples in a deliberately chosen distance from the current reference solution. The scientific and practical value of the baseline method [Zhang et al., 2020] is all but clear. The current paper improves upon that baseline by getting rid of two tuning parameters, which is a nice contribution.
In other words: The contribution of the paper in itself is valuable, provided that the baseline method is. I find the second part near-impossible to judge from the provided material. Why should a bunch of 1D estimators be vastly superior to a vector estimator? Looking for an answer I checked [Zhang et al., 2020]. However, there is no unbiased comparison with competitors, since all results are subject to extensive problem-specific hyperparameter tuning.


THE PROPOSED METHOD

Section 2 provides a motivation of the method of [Zhang et al., 2020]. There is not much detail, in particular no pseudo code, and the various tuning parameters are missing.

The proposed method is based on line search. The search itself replace the learning rate parameter, which is then used with with an exponentially fading record technique to adapt the sampling radius.

When setting the algorithm parameters, it is assumed that there is a bounded domain inside of which optimization is performed. However, at least conceptually, all test functions are defined on unbounded domains (real vector spaces). In my opinion, a solid optimization algorithm should not depend (critically) on the presence of bounds. Gradient descent does not, so why should a line search method?

Also, the default target precision is quite bad: L_{min} = L_{max} / 200, or 0.5% of the diameter of the space. In a 1000D unit hypercube that's roughly 0.158. I don't think that's a satisfactory precision. An evolution strategy will happily identify the optimum to arbitrary precision, and close to numerical precision in practice. Instead of limiting the algorithm in such a way it should feature a truly adaptive mechanism.

I simply cannot believe the condition $|F(x_t) - F(x_{t-1})| / |F(x_{t-1})| < \gamma$. What prevents us from dividing by zero? The condition seems to assume that the optimal value is zero. That's a no-go.

Under "reset the smoothing radius" it is claimed that the method "converges" in less than 10 optimization steps. How comes? Did I overlook a second order method? No first order method ever converges that fast, unless the problem is completely trivial. Anyway, the very need for a reset mechanism and the absence of a detection/trigger mechanism is a bad sign.

Random generation of the rotation matrix: It should be said that sampling a random orthogonal matrix from the uniform distribution is a cubic time procedure, which is surely undesirable. On the other hand, generating a fresh matrix from time to time is obviously a good idea.


AN IMPORTANT (SIDE) NOTE ON EVOLUTION STRATEGIES

The paper includes many weak and even clearly wrong statements about ES. I see lots of misconceptions, not only in this paper. Somehow, when it comes to ES, the ML community seems to live in a parallel universe; it really started to invent its own reality. ES research does not (primarily) take place at ML conferences, but at GECCO, FOGA, PPSN and a few others. However, nobody in ML seems to read any further than Salimans plus a few (good) papers published in JMLR. When talking about ES, ignoring these conferences simply does not work. Nearly none of the algorithms discussed in ML actually qualify as ES, according to the very definition of the method. More often than not, there is no ranking (and selection) of solutions, and no step size control, which was at the core of ES since their inception nearly 50 years ago. The NES family of algorithm was exploring an interesting connection to the gradient-based world. However, concluding that ES are gradient estimators is just wrong.

Here is a concrete point how this attitude affects the current paper. Appendix B.2 states verbatim: "ASEBO refers to Adaptive ES-Active Subspaces for Blackbox Optimization proposed in (Choromanski et al., 2019). This is the state-of-the-art method in the family of ES."
Wow...what? I must have lived behind the moon, I missed the new SOTA. That statement is shared by exactly no single lead figure in evolution strategies research. Maybe it was made up by the authors, or claimed by the cited paper or github repository -- I don't know, and it is anyway useless to track down its origin. In my surprise, I checked the ASEBO repository and the paper cited therein. There the ASEBO algorithm was found to be superior to several variants of CMA-ES. Knowing the compared algorithms well, none of the experimental results on analytic benchmarks looked even remotely plausible to me. I quickly fired up a control experiment on the 1000-dimensional sphere, where ASEBO was reported to reduce the function value by about a factor of two in 20K function evaluations. With identical setup, the three ES variants I tested reduce it by a factor between 30 and 100, with zero tuning, including an 11-lines (1+1)-ES. My test also included LM-MA-ES, which was included in that benchmark, where it did not do anything.
Is it just me? This reminds me of "alternative facts". It is no wonder that ML research is chasing its own tail.

Citing again from the paper: "A exemplary type of these methods is Evolution Strategy (ES) based on the traditional GS, first introduced by (Salimans et al., 2017)."
The term "evolution strategy" goes back to Ingo Rechenberg's PhD thesis, published in the 1970es. The state-of-the-art is marked by CMA-ES and its variants. For a very brief history check wikipedia: https://en.wikipedia.org/wiki/Evolution_strategy). NES plays only a very minor side role in the development.
In 2017, Salimans was standing on the shoulders of giants. There exists a vast (and in parts old) literature on evolution strategies, including very relevant but rarely cited work on applications in RL. The OpenAI paper simply leverages that literature, in poor a way, and without citing it properly (maybe that's where the problem started). The algorithm described therein has the same extremely slow convergence speed as pure random search (due to a lack of step size control), unless it is combined with ADAM, which is apparently done in the experiments but not even stated.

A different point -- the paper
Ollivier, Yann, et al. "Information-geometric optimization algorithms: A unifying picture via invariance principles." The Journal of Machine Learning Research 18.1 (2017): 564-628.
describes the gradient flow of proper ES, which are based on ranking of solutions. This flow is defined on the statistical manifold of search distributions, not directly on the search space. There is really a lot more behind evolution strategies than what was put forward in the Salimans paper. In particular I argue that ES should not be listed under the heading "Zeroth order methods based on local gradient surrogate". The gradients estimated by ES have never been local.

A final point, highlighting the often shallow understanding of methods like CMA-ES: The covariance matrix has a quadratic number of coefficients in the problem dimension. For successful adaptation, it requires roughly a quadratic number of data samples -- that's how its learning rates are tuned. Anyway, they are not well tuned for problem dimensions much larger than 100. When running an experiment on a 10^3-dimensional problem this means that we need a function evaluation budget in the order of 10^6 for the technique to have a meaningful effect. When running for 20K evaluations, CMA does not make sense. Just drop the CMA technique and save 99% of the computation time. Or better, use a low-rank approach like LM-MA-ES. The method is even cited [Loshchilov et al., 2019], but then ignored.


EXPERIMENTS

It is quite clear that the experiments are not designed to answer any specific research question. Instead, there is a battery of generic performance tests, with the single focus of being best in class. That's unreasonable. For example, it cannot be expected that the new method outperforms the same method with optimal learning rate schedule. It may still be considered superior if it comes close, removes two tuning parameters, and the adaptation mechanism is robust.

I appreciate the code supplement. However, it does not allow me to reproduce the experiments since it does not include the competitor methods. Indeed, I quickly implemented a CMA-ES control experiment straight into the provided code, using the ellipsoid benchmark. The performance curve I get looks VERY different from what is reported. It is initially MUCH faster and then slows down when it needs to adapt the covariance matrix.
I also ran a few control experiments, with my own code. I picked the Rosenbrock function as an example. Critical information is missing in the paper, like the initial step size. No matter what I do, I fail to reproduce the plots, even qualitatively. The only way to make CMA-ES stall completely for 5000 evaluations is to start with a too small step size.

Line search is not a new technique, and it was applied to ES for step size control. The resulting "Two-point adaptation" technique is the maybe second-best step size adaptation rule for CMA-ES. Here is a link to the original technique, which was refined over the years:
https://arxiv.org/abs/0805.0231
With the small function evaluation budget (compared to the problem dimension) used in all experiments, results are dominated by initialization and step size control. Therefore please compare to this method.

This paper claims to provide a well working algorithm, however, it provides very few experiments. The ES community has an established standard benchmark suite: https://github.com/numbbo/coco. It includes a "high-dimensional" problems track, and it delivers far more insights into strengths and limitations of algorithms than the presented material.

A simple but very important test problem is missing: the sphere function. I would really like to see the performance on this problem, since it is directly related to the quality of the gradient estimator. With a perfect gradient, an exact line search solves the problem in a single step. However, does that happen? If not, what is the limiting factor -- the gradient or the line search? The sphere would be a good opportunity for a deeper investigation and real insights.

The most important experiments are missing! The contribution of the paper is to remove tuning parameters from Zhang's method. Therefore I'd like to see the effect of the new automatic choice, with manually tuned (optimal) and naive parameter choices as baselines. Does the new mechanism work close to optimal or not? That's the core question that needs to be answered.

The way the numerical results are presented is highly problematic. On smooth unimodal problems, any well-designed ES based on ranking of solutions exhibits convergence at a linear rate. The better the rate, the faster does the algorithm converge asymptotically. This rate is the slope of the graph of $\log(f(x_t) - f(x^*))$ as a function of $t$ (time). Therefore, in all serious ES research, performance plots ALWAYS use a log-scale for the vertical axis. I would go so far to say that the presented plots are close to meaningless.

Table 1: Please plot progress curves instead of an arbitrary single point in time (1500 function evaluations).

Section 4.3 reads as if super mario level generation would be a widely used benchmark. Then why does the (actually very nice) paper by Volz et al. have only 11 citations on Google scholar?
Also, citing a youtube video for the A-star Mario agent is unhelpful. I'd like to see the method or the code or both, but not a demo. This is supposed to be research, not entertainment. Also, that agent is a hopelessly over-optimistic definition of a "playable" level.
There is problem-specific hyperparameter tuning for this task, and only a single baseline method tested, for a shady reason. Please provide a complete comparison to all baselines and avoid problem-specific tuning. This is black-box optimization. If you tune then it counts towards your function evaluation budget.

Appendix A, description of the Rosenbrock problem: "The ridge changes its orientation d - 1 times."
The function is a 4th order polynomial. The ridge is parabola-shaped. There are no discrete changes whatsoever. Anyway, the function evaluation budget is so low that no algorithm even gets into the mode of really following the ridge, and approaching the optimum is completely impossible. In my opinion, even using the benchmark in this regime is misleading.


CONCLUSION

The proposed method may or may not be of value, I cannot really tell. The most relevant comparison experiments are missing. Furthermore, frankly, after my own experimentation, I do not have trust in the results. Also, I need to see BBOB/COCO benchmark results, including baselines.
And finally, please correct various statements on ES.


RECOMMENDATION

The paper has many weaknesses. Most prominently, the experimental evaluation is problematic at least, and the most important experiments are missing. Therefore I recommend to reject the paper.

---

> ### Author Response · Authors · 2020-11-18
> **Response to AnonReviewer2**
>
> We thank the reviewer for spending so much time on our paper. Please see the general response for our rebuttal regarding to the contribution of our work.
>
> THE PROPOSED METHOD
>
> “Section 2”: Due to the page limit, we only provide a brief review of DGS in Section 2. We refer the interested reader to [Zhang et al., 2020] for full details.
>
> “dependence on the presence of bounds”: For a nonlocal exploration method, we believe it is reasonable to take advantage of the size of the search domain. This information is merely used as an upper limit for searching range and smoothing radius. Gradient descent method is a local method, so does not use this information, but the presence of bound is also helpful for methods like CMA-ES in initializing exploration radius.
>
> “default target precision”: We should have mentioned that this default target precision is only in effect in low-dimensional optimization. In high-dimensional problem, since we can choose a large number of points along the searching direction, we do not need to set $L_{min}$ a priori: the minimum range will be very small anyway. For example, in our 1000-D benchmark function testing (Section 4.1), $S$ is set to be 200 (5% of samples needed to compute DGS gradient). We choose contraction rate $\rho = 0.9$ (conservative for line search) and the minimum range is $0.9^{200} * L_{max} = 4.4*10^{-8}$. Thank you for pointing this out.
>
> “condition $|F(x_t)−F(x_{t−1})|/|F(x_t−1)|<\gamma$”: Sure. We added 1e-10 to the denominator to avoid division by zero.
>
> “it is claimed that the method "converges" in less than 10 optimization steps”: Converge is not a correct word. Thank you for pointing it out. What we meant is that within 10 steps (40000 function evaluations), AdaDGS already reach function values lower than its competitors can at the end. In this sense, AdaDGS outperforms other baseline without resetting radius.
>
> “Random generation of the rotation matrix”: We randomly rotate and translate the function at the beginning of each trial to make the optimization problem harder and demonstrate that AdaDGS does NOT exploit separability of the test function for its performance.
>
> NOTE ON EVOLUTION STRATEGIES
>
> Thank you for your careful note on Evolution Strategies. We are definitely aware that ES starts in 1970s and has a vast literature. We appreciate with your nice remarks but also clarify that our comments on ES in our paper were really just meant for the line of ES based on Gaussian smoothing formula, starting with (Salimans et al, 2017). In that context, we hope our statements are more reasonable. In any case, we will edit to make this clear in the next revision.
>
> EXPERIMENTS
>
> “the experiments are not designed to answer any specific research question”: Our goal was to perform a comprehensive comparison of AdaDGS and its competitors in a wide variety of scenarios: unimodal, multimodal, with/without global structure, high- and medium-dimensional, etc. In fact, AdaDGS is most advantageous in optimizing high-dimensional, multimodal landscape with global structures, but the experiments were not just focused on that case.
>
> “CMA-ES experiments”: We re-ran the CMA-ES test and updated our paper with new error plot. The initial step size is set to be ¼ of the search domain width, as noted in the Appendix.
>
> “Two-point adaptation”:  Thank you for your suggestion. We will consider comparison between AdaDGS and CMA-ES-TPA in future research.
>
> “coco suite”: We agree that more benchmark functions could be tested. However, the presented test set also covers a wide variety of landscapes and should provide some insights into strengths and limitations of algorithms. In the future, we will consider performing AdaDGS on COCO test suite. Thank you for your suggestion.
>
> “sphere function”: We include the sphere function test to the revised paper. Our method converges in 2 iterations.
>
> “performance plots in a log-scale”: Several test functions in our paper are so challenging that none of the tested methods can reach global minima.  We believe evaluating the performance of optimization methods based on how much the loss function value is dropped is totally reasonable. In fact, performance plots in linear scale for these benchmark tests are commonly seen. As suggested by the reviewer, we include plots in log scale in the Appendix.
>
> “Mario level generation”: We believe it is reasonable to tune an adaptive algorithm slightly when transferring across different problem classes as long as: i) the tuning is cheap (here, only a few runs to determine the maximum and minimum range of line search); ii) the hyperparameters are then fixed in all problems within the new class. However, we perform and show results with same hyperparameters used in the benchmark function tests as suggested.
>
> “Rosenbrock problem”: Certainly, no algorithm can approach the global optimum. However, it still makes sense to compare how fast different methods can drop the function value, in our view.

---

### Official Review · AnonReviewer1 · 2020-10-26
**Lacking evaluation of the primary contribution (the adaptive mechanism)**

**Rating:** 4
**Confidence:** 4

**Review:**

** Summary **

This paper considers the problem of adapting the learning rate and smoothing parameter in the Directional Gaussian Smoothing (DGS) algorithm. The authors claim DGS is particularly sensitive to these parameters and thus attribute the strong experimental results to these changes.

** Primary Reason for Score**

The sole contribution of this work is an online hyperparameter optimization approach to improve DGS. The comparisons vs. other methods are strong, but there is no comparison vs. DGS/other approaches to learn hyperparameters for DGS. That should be the focus here, since the contribution of this paper is solely the adaptive mechanism. It would be useful for example to know if this method is able to learn the best values without any prior information. That may make the use of backtracking line search more prominent across other methods, as if it is just useful for DGS then it may have limited impact.

** Strengths **

1) The topic of learning hyperparameters on the fly is well-motivated and an important direction to improve many existing methods.
2) The experiments are interesting, with a wide range of tasks considered. The authors use a significant number of seeds, and present the results clearly.
3) The discussion of performance on different tasks (in particular, Section 4.1) is useful.
4) The authors included several (challenging) baselines to demonstrate the performance.
5) The proposed idea of including dimensionality reduction techniques seems like interesting future work.

** Weaknesses **

1) It is not clear how much improvement the adaptive component makes vs. DGS. It would be great to see detailed ablation studies of the individual hyperparameter optimization methods and whether they learn optimal configurations without prior knowledge. For instance, it is entirely possible that DGS performs better than AdaDGS and the reader wouldn’t know. That is the minimum that should be included however, ideally we would be able to understand where the gains come from (assuming they exist).
2) DGS itself is not published (as far as I can see) and has only been cited once. Therefore, it is hard to see significant impact for this work, as it is presented as a method to improve *just* DGS. It would be interesting to see if the proposed method can improve other algorithms too.
3) It would be useful to get a comparison vs. other hyperparameter optimization methods, since that is the sole contribution of the work. This has been studied extensively (for example, online-learning approaches or blackbox methods like PBT/Hyperband).
4) Section 4.3 is a nice application of blackbox optimization methods, however, given the noise in Figure 3, it is hard to be convinced that AdaDGS is any better than IPop-CMA.
5) We don’t get to see how the new meta-parameters were chosen. It is alluded to on p4, for example: “We do not observe significant benefit of using more than 5 GH quadrature points per direction. In some tests, 3 GH quadrature points per direction are sufficient.” but we never see this. Given the hyperparameters for the baselines were likely taken off the shelf, it is possible that AdaDGS was simply tuned better for the tasks.

** Minor issues **

- It would be useful to include a wall-clock comparison, if possible.

---

> ### Author Response · Authors · 2020-11-18
> **Response to AnonReviewer1**
>
> We thank Reviewer 1 for the constructive comments and suggestions. We address the reviewer’s concerns below.
>
> 1.	**Comparison with non-adaptive DGS:** The reviewer was absolutely right. We will include in the revised paper a comparison between the performance of AdaDGS with non-adaptive DGS. The adaptive component makes a clear improvement over non-adaptive DGS.
>
> 2.	**Impact of this work:** Please see the general response for our rebuttal regarding to the contribution of our work.
>
> 3.	**Comparison with other hyperparameter optimization methods:** Thank you for your suggestion of PBT/Hyperband method. Our approach represents an adaptive technique to update the hyperparameters along the run, while hyperparameter optimization methods often conduct an extensive search on hyperparameter space for their optimal values. We will consider to compare AdaDGS and DGS equipped with hyperparameter optimization in future research.
>
> 4.	**Performance of AdaDGS and IPop-CMA in Section 4.3:** In Figure 3, we show AdaDGS outperforms Ipop-CMA in three tests (MaxSkyTiles, MaxKills and MaxJumps) and is slightly underperform in one another (MaxEnemies). In Figure 4, we show a set of the Mario levels generated with MaxSkyTiles objective. Statistically, the levels generated by AdaDGS have higher number of sky tiles than those from Ipop-CMA (i.e., users see pattern Top-Left more frequently and Bottom-Left less frequently). The “noise” in the levels is part of the generative model we used.
>
> 5.	**How the new meta-parameters were chosen:** Please see general response for our clarification of choosing hyperparameters. Actually, AdaDGS does not simply tune hyperparameters better for the tasks, but, via a line search procedure, replacing the sensitive hyperparameters of baseline DGS by new hyperparameters which is intuitive and not sensitive. In Section 4.1 and 4.2, we use 5 GH quadrature points, but in Section 4.3, 3 GH quadrature points per direction is enough.

---

### Official Review · AnonReviewer4 · 2020-10-28
**An incremental work of DGS (Zhang et al., 2020))**

**Rating:** 4
**Confidence:** 4

**Review:**




In this paper, the authors apply a line-search of the step-size parameter of DGS (Zhang et al., 2020)) to reduce tunning. A heuristic update rule of the smooth parameter in DGS (Zhang et al., 2020))  is also used.  Overall, I think it is an incremental work of DGS (Zhang et al., 2020)).  The contribution is too marginal.


Pros

1. The paper is well written and well organized.

2. Twelve synthetic functions and two practical problems are evaluated.  The set of synthetic functions covers the most character of multi-model problems and is suitable for the evaluation of the performance of AdaDGS on multi-model problems.



Cons.

1.  There are several hyperparameters of the proposed AdaDGS, e.g., L_{min}, L_{max},  S,  and gamma.  How to choose these hyperparameters?  Does AdaDGS sensitive to these hyperparameters?

2.  In the experiments on synthetic problems, the initialization point and optimal point are not clear. What is the x_{opt} for each synthetic problem?  What is the initialization point for each method?  Actually, the optimization performance depending on the distance between the initialization point and the optimal point. It is challenging for problems with a large distance. For example, the authors can check the performance on rotated Ackley and Rastrigin with x_{opt} = 100* ones(d,1) and x_{ini} = 0 .  I would like to see a comparison with other baselines on problems with increasing distance || x_{opt} - x_{ini} ||. The authors can fix x_{ini} at zeros, and set x_{opt} = 2*ones(d,1) ,  x_{opt} = 5*ones(d,1) and x_{opt} = 10*ones(d,1).

3.  For high-dimension multi-model problems, a large population size can reduce stuck at bad local optimum at the early phase.  In the experiments, the population size of CMA-ES is different from AdaDGS. Keep the population size the same can reduce the influence of this factor.  I would like to see a comparison with CMA-ES with a population size the same as AdaDGS.

4.  In the experiments, the dimensions of synthetic problems are thousands.  The regime of dimensions of problems that are suited for AdaDGS is not clear.  I would like to see a comparison with baselines on  100-dimensional problems.


Kindly reminding: the template may be inappropriately used. It is not “Published as a conference paper at ICLR 2021”.

---

> ### Author Response · Authors · 2020-11-18
> **Response to AnonReviewer4**
>
> We thank Reviewer 4 for the constructive comments and suggestions. Please see the general response for the contribution of our paper. We address the reviewer’s particular concerns below.
>
> 1.	**Hyperparameter selection:** The main goal of our paper is to develop a new adaptive strategy to remove the need of hyperparameter fine tuning, which hinders the performance of DGS method. In the work of [[Zhang et. al., 2020]](https://arxiv.org/pdf/2002.03001.pdf), there are two parameters that require careful selection - the learning rate and smoothing radius. While our adaptive approach still relies on some hyperparameters, AdaDGS is not sensitive to these hyperparameters. In particular:
> -	$L_{min}$ and $L_{max}$ represent the minimum and maximum possible step, and $S$ is the number of points to be evaluated in the line search. The rule of thumb is: the smaller $L_{min}$ is and the bigger $L_{max}$ and $S$ are, the better. One advantage of the line search strategy in high dimensional optimization is that the sampling cost for line search is very small compare to that of evaluating DGS gradient (only 5% in our synthetic function test). We found the recommended values lead to competitive results in all of our tests, but of course, more conservative choices of $L_{min}$, $L_{max}$ and $S$ (i.e., more aggressive line search) are also good and do not affect the total computational cost much.
> -	$\gamma$ measures the relative decreasing rate between two consecutive iterations and is used to detect potential stagnation at local minima, in which case we reset the smoothing radius. $\gamma$ should be chosen small, but otherwise, our method is not sensitive to $\gamma$.
>
> Additional discussion on our hyperparameter choice can be found in the general response.
>
> 2.	**Initialization:** We included the value of $x_{opt}$ for standard synthetic functions in the revised paper (see Appendix A). Our tests were indeed performed for varied distance between the initialization point and the optimal point. More specifically, in each trial, we randomly generated $x_{ini}$ in initial search domain, say, $[-1,1]^d$, and applied a random translation to the test function, such as the $x_{opt}$ lies in a subdomain with width being $0.8$ of the original width, say $[-0.8,0.8]^d$. With this setup, we allow the distance $||x_{opt} - x_{ini}||$ to be anywhere from $0$ to $0.9$*(domain diagonal).
>
> 3.	**Population size of CMA-ES:** We employed the population size $4+3*\log(dim)$, suggested by CMA-ES’ authors, e.g., [Auger, Hansen 2005] and widely used in literature. We believe CMA-ES performs better with this choice than with very large population size.
>
> 4.	**Regime of dimensions for AdaDGS:** Our method is best-suited to optimize high-dimensional, multimodal functions (examples are 1000D - 6000D synthetic functions in our paper). But this is not exclusive. We showed two medium-dimensional problems (15D and 32D), where AdaDGS can also produce competitive results.
>
> ---
> [Auger, Hansen 2005] A. Auger and N. Hansen. A restart cma evolution strategy with increasing population size. In2005IEEE Congress on Evolutionary Computation, volume 2, pp. 1769–1776 Vol. 2, 2005

---

### Author Response · Authors · 2020-11-18
**General response**

We thank the reviewers for their helpful and constructive feedback. We first address the general concerns raised by multiple reviewers and then summarize the changes made in the revised draft.

**Contribution of our work.** Our main contribution is developing an adaptive mechanism for optimization algorithm using DGS gradient, which removes the need of hyperparameter fine tuning. We believe that the hyperparameter tuning is a major hurdle in optimization, and for any new optimization method to attract users and expand its applicability, this problem worth serious consideration.

DGS is a recent optimization method built upon on a novel nonlocal gradient for long range exploration, best suited to optimize on a multimodal landscape. Clearly, the impact of this paper would depend on that of baseline DGS. Although [[Zhang et al., 2020]](https://arxiv.org/pdf/2002.03001.pdf) has not been officially published, we believe extensive comparison conducted in therein shows convincing potential of this approach. Certainly, the selection of hyperparameter therein was non-intuitive and suboptimal. That is exactly the motivation driving the development of this work.

The reviewers were completely right that our paper lacks evaluation of the adaptive mechanism. To make this clear, we added result of the baseline DGS approach in the revised paper, so the reader can clearly see how much we gain with the adaptive mechanism.

**How do we choose the hyperparameters.** As oppose to baseline DGS, our AdaDGS algorithm is not sensitive with the newly introduced hyperparameters, which is selected intuitively. Three of the parameters, $L_{min}$, $L_{max}$ and $S$, just determine how aggressive we want to conduct the line search. Our method is best suited for high-dimensional optimization. Here, the key advantage is that the budget for line search is very small compared to that of computing DGS gradient, so we can approximate the optimal learning rate from a generous number of function evaluations along DGS direction. The default value of $S$ mean that we spend 5% of DGS computational cost to conducting line search. Certainly, a more aggressive line search by extending the range (decreasing $L_{min}$ and increasing $L_{max}$), increasing the point to be visited with line search ($S$) can only lead to even better result. However, the default values of these parameters prove to be adequate throughout our test. In any case, if the user opts to be more aggressive with line search, they can do so by even doubling or tripling the number of points to be visited along DGS direction. The total number of function evaluations will increase by only 5% and 10% correspondingly.

$M$ is the number of Gaussian Hermit points in each direction, set to be $5$. We observe no significant improvement with $M>5$.

Our choice of $\sigma_0$ and $\gamma$ is quite trivial. We choose initial smoothing radius $\sigma_0$ to be the width of the search domain, so that it agrees with the maximum allowable searching range. This is based on our intuition that these two quantities should be closely related. $\gamma$ represents the relative decreasing rate tolerance to trigger the smoothing radius reset, so naturally being small. We set $\gamma = 10^{-3}$.

**Changes.** We highlighted modifications in the revised paper in red.
1.	Experiment results for baseline DGS are added.
2.	Test results for sphere functions are added.
3.	CMA-ES results are updated for benchmark functions examples.
4.	Performance plots in log-scale are added to the appendix.
5.	Several clarifications and edits as noted in the responses.

---

### Decision · Program_Chairs · 2021-01-07
**Final Decision**

**Decision:**

Reject

**Comment:**

The main goal of this paper is to develop a new adaptive strategy to remove the need of hyperparameter fine tuning, which hinders the performance of DGS (Zhang et al., 2020) method. This paper applies a line-search of the step-size parameter of DGS  to reduce tuning.  A heuristic update rule of the smooth parameter in DGS (Zhang et al., 2020)) is also used.

Pros
+ The topic of learning hyperparameters on the fly is well-motivated and an important direction to improve many existing methods.
+ A wide range of tasks are considered in the experiments are interesting, with a wide range of tasks considered.


Cons
- Reviewers have found the contribution to be incremental and marginal. Specifically, the reviewers have expressed concerns on the impact of the work since it verifies improvement upon DGS only. The paper could be stronger by showing evidence of improving other algorithms.
-  In the work of [Zhang et. al., 2020], there are two parameters that require careful selection - the learning rate and smoothing radius. However, the proposed approach still relies on some hyperparameters. Although the authors claim that the method is not sensitive to these hyperparameters, it could be better justified.
- The initial version lacks evaluation of the adaptive mechanism, although the authors added the comparison in the revised version.
- The paper could be further improved by comparing against other hyperparameter optimization methods.

We acknowledge the detailed response and the modifications of the manuscript. We believe the paper will make a more profound contribution and impact after addressing some of the major concerns raised by the reviewers.